# Anthropometric Profiling and Changes in Segmental Body Composition of Professional Football Players in Relation to Age over the Training Macrocycle

**DOI:** 10.3390/sports11090172

**Published:** 2023-09-05

**Authors:** Wiktoria Staśkiewicz-Bartecka, Elżbieta Grochowska-Niedworok, Grzegorz Zydek, Mateusz Grajek, Agata Kiciak, Agnieszka Białek-Dratwa, Ewa Niewiadomska, Oskar Kowalski, Marek Kardas

**Affiliations:** 1Department of Food Technology and Quality Evaluation, Department of Dietetics, Faculty of Public Health in Bytom, Medical University of Silesia in Katowice, Ul. Jordana 19, 41-808 Zabrze, Poland; akiciak@sum.edu.pl (A.K.); mkardas@sum.edu.pl (M.K.); 2Department of Health Sciences and Physical Culture, University of Applied Sciences in Nysa, Ul. Ujejskiego 12, 48-300 Nysa, Poland; elzbieta.grochowska-niedworok@pwsz.nysa.pl; 3Department of Sport Nutrition, Jerzy Kukuczka Academy of Physical Education in Katowice, Ul. Mikołowska 72A, 40-065 Katowice, Poland; g.zydek@awf.katowice.pl; 4Department of Public Health, Department of Public Health Policy, Faculty of Public Health in Bytom, Medical University of Silesia in Katowice, Ul. Piekarska 18, 41-902 Bytom, Poland; mgrajek@sum.edu.pl; 5Department of Human Nutrition, Department of Dietetics, Faculty of Public Health in Bytom, Medical University of Silesia in Katowice, Ul. Jordana 19, 41-808 Zabrze, Poland; okowalski@sum.edu.pl; 6Department of Epidemiology and Biostatistics, Faculty of Public Health in Bytom, Medical University of Silesia in Katowice, Ul. Piekarska 18, 41-902 Bytom, Poland; eniewiadomska@sum.edu.pl

**Keywords:** anthropometric profiling, body composition, football players, macrocycle, segmental body composition

## Abstract

Body composition is an important indicator of the overall health and fitness of team sports athletes, including in football, and therefore, anthropometric profiling of elite football players is useful as part of determining their skills, strengths, and weaknesses to develop effective strength and conditioning programs. One of the tools available to coaches to track correlates of performance and health is routine body composition assessment. The purpose of this study is to describe and compare the body composition and anthropometric profiles of players using the Direct Segmental Multi-Frequency Bio-Electrical Impedance Analysis method, and to manage body composition throughout the round in the 2020–2021 season. The investigation was carried out during the Polish football league, PKO BP Ekstraklasa, spring round of the football season 2020–2021, in which male football players participated. Athletes between the ages of 18 and 25 (*n* = 16) made up the younger age group, while those between the ages of 26 and 31 (*n* = 22) made up the older age group. This manuscript is a continuation of the presentation of the results of the study, which was conducted between 7 January and 23 July 2021. At different stages of the macrocycle, participants underwent six different body composition analyses. The younger and older groups of athletes were compared, as well as measurements of time points 1–6. The dominant extremities, assisting extremities, and trunk had larger fat-free mass contents in the older age group. In the study groups, there was a difference in the fat-free mass content between measures 1–6 that was statistically significant. In the younger group, there was a statistically significant difference in the amount of fat mass content between measurements 1–6. In the older age group, no statistically significant changes were found. The study showed changes in fat-free mass and fat mass in body segments; differences were observed between age groups and between different moments of measurement. Age is an important factor in determining body composition and is also related to an athlete’s experience and seniority. Anthropometric profiling and comprehensive body composition analysis are important tools used in preparing athletes for competition.

## 1. Introduction

Profiling elite athletes is an effective approach to pinpoint abilities, and it is essential for identifying the strengths and shortcomings of athletes, as well as for creating effective strength and conditioning regimens [1]. In team sports studies, athletes have frequently reported larger total body mass, fat-free mass, and bone mass, along with lower total body fat mass, when compared to their age-matched control counterparts across a range of competition standards and age levels [2]. The older group’s lengthier training history has been proposed as the most likely reason for the reported discrepancies since the number of seasonal exercise sessions completed by the older and younger athletes was roughly the same. Although it is still up for debate whether a certain body type predicts success at the professional team sports level, few studies have been conducted to show variations across groups over the course of the training macrocycle [3,4].

Training should be considered a process that technically, tactically, psychologically, physiologically, and physically prepares an athlete to achieve the highest possible level of performance [5]. The training process is related to improving performance, so it must provide adequate stimulus for adaptation; adequate means of assessing progress (monitoring changes); and other relevant measures, including rest and recovery, mental support, proper nutrition, and properly selected supplementation [6]. One measure to assess performance progress is body composition assessment.

Cyclicality and periodization of training are now unquestionable in the training process, their meaning deriving from the construction of the sports form and the factors related to its formation: the undulating tendency of increasing training loads, the necessity of alternation of work and rest phases, and the order of training accents [7]. In football, one cycle is a six-month macrocycle, called the spring round, lasting from January to June, and the fall round from July to December. In one round, three periods are distinguished: preparatory, competitive, and transition [8]. The different periods of the macrocycle are successive stages of managing the development of the sports form [9]. The preparatory period in the discipline of football is aimed at developing fitness in preparation for the starting period [10]. The competitive period is about maximizing sports performance during competition, based on the fitness and skill dispositions obtained during the preparatory period [8]. The transition period is characterized by a complete cessation or significant reduction in participation in training. During this period, athletes may be involved in recreational sports activities or a significant reduction in training participation [11]. This period should be viewed by athletes as an opportunity to recuperate before the next season. A complete lack of training stimuli can adversely affect the physical performance and physiological parameters of football players [12].

Body composition is an important indicator of athletes’ physical fitness and overall health [13]. In athletes, body composition is not only considered a condition of health, but also influences athletic performance by affecting movement quality and performance levels. In recent years, anthropometric and fitness characteristics have been identified that predispose some athletes to success in football [14].

Currently, no study has described the body composition and anthropometric profiles of players in Poland’s top PKO BP Ekstraklasa league. Therefore, the purpose of this study was to describe and compare the body composition and anthropometric profiles of players and manage body composition throughout the round in the 2020–2021 season using the Direct Segmental Multi-Frequency Bio-Electrical Impedance Analysis (DSM-BIA) method.

## 2. Materials and Methods

### 2.1. Study Design

The inquiry was conducted between 7 January and 23 July, 2021, during the PKO BP Ekstraklasa spring round of the 2020–2021 football season—the top-tier Polish football league. The research group consisted of male football players from two Silesian clubs participating in the PKO BP Ekstraklasa 2020/2021. This manuscript is a continuation of previous work, in which the used methodology has been described previously [15,16].

The inclusion criterion for participation in the study was to be a professional football player in one of the clubs involved (a player having signed a professional contract), agree to participate in the study, and not have any long-term injuries that would prevent them from participating in training and games for the duration of the seven-month study. Exclusion criteria were [15,16] transferred from one team to another during the study period; absence from practice and competition lasting at least 14 days due to illness, injury, isolation, or quarantine as a result of the COVID-19 pandemic; and failure to participate in at least 1 of the 6 measurements for any other reason than those mentioned above.

The Declaration of Helsinki of the World Medical Association guided the conduct of this study. The study protocol (PCN/0022/KB/68/I/20) was reviewed by the bioethics committee of the Silesian Medical University in Katowice and approved. All participants provided written, fully informed permission to participate in the study.

During the pre-competitive, competitive, and transitional phases of the spring 2020/2021 PKO BP Ekstraklasa football season, six body composition examinations were performed. The timing of measurement days is determined by league and cup competitions, as well as by allowing adequate time for post-workout recovery (according to the test protocol advised by the InBody producer, the gap between body composition analysis and physical activity contributed a minimum of 24 h). Measurements were made at the following macrocycle timepoints, as defined previously. The 1st measurement was made before the start of the preparatory period; the 2nd measurement after the preparatory period; the 3rd and 4th measurements during the competitive period; the 5th measurement after the competitive period; and the 6th measurement after the transition period [15,16].

### 2.2. Participants

The 58 football players included in the study were between the ages of 18 and 31. Athletes from different nationalities participated in the study: Polish, Slovakian, Spanish, Portuguese, Greek, Slovenian, Czech, Austrian, Danish, Hungarian, Ghanaian, and Gambian.

Athletes between the ages of 18 and 25 (*n* = 16) made up the younger age group, while those between the ages of 26 and 31 (*n* = 22) made up the older age group. The ideal age for the growth of the highest athletic performance and the accomplishment of full ontogenetic development were taken into consideration when creating this division [14,17,18,19]. A total of 38 athletes provided 228 measurements of their body composition, and the inclusion and exclusion criteria were taken into account.

### 2.3. Procedure

Before the first body composition study, height was assessed. Body mass was determined for each body composition measurement.

Height (cm) and body mass (kg) were measured with 0.1 cm and 0.1 kg precision (SECA 756, Seca gmbh & Co. KG., Hamburg, Germany) using the InBody 770 (InBody USA, Cerritos, CA, USA). The Body Mass Index (BMI) was calculated by dividing the body mass (kg) by the square of the height (m). The results were used as the basis for comparing height-to-weight ratios for the European population to WHO (World Health Organization) recommendations and guidelines [18,20].

Body composition was determined using the Direct Segmental Multi-Frequency Bio-Electrical Impedance Analysis (DSM-BIA) method (InBody 770, InBody USA, Cerritos, CA, USA). DSM-BIA, which is based on the idea that the human body is made up of five linked cylinders, analyzes impedance directly from particular internal compartments. Using a tetrapolar eight-point touch electrode system, the test subject’s torso, arms, and legs are each separately tested for impedance at six distinct frequencies (1 kHz, 5 kHz, 50 kHz, 250 kHz, 500 kHz, and 1000 kHz). The analyzer operates on an 80 uA current [21,22].

To analyze the results, anthropometric measurements were classified according to lateralization. The classification was based on the participants’ declaration of which limb was the dominant or assisting limb. The dominant limb was indicated as the one with greater strength and motor skills.

With the help of Lookin’Body Software version 120.3.0.6, the body composition parameters were discovered. Measurements were calculated by a predetermined methodology and all manufacturer’s guidelines. Before each measurement session, the analyzer was checked using a calibration circuit with known impedance (resistance = 500.0; reactance = 0.1; error = 0.9%). The research adhered to standard procedures and included pertinent literature. The tests were performed according to a standard protocol, as recommended by the manufacturer of the device: fasting or 2–3 h after a meal, at a fixed time of day, after a bowel movement, at least 24 h after the end of intense physical activity, without shoes or socks, in underwear, with clean and dried feet and hands without applied cream or lotion [21,22,23].

### 2.4. Statistical Analysis

Data were analyzed using Statistica 13.3 (StatSoft Polska Sp. z o.o., Cracow, Poland) and the R 4.0.0 package (2020) (The R Foundation for Statistical Computing, Vienna, Austria), in accordance with the GNU GPL license. The data were summarized as the mean and standard deviation.

The Shapiro–Wilk test was used to assess the conformity of a normal distribution. The significance of the differences between the means in the age groups (19–25 years, 26–31 years) was assessed using the Student’s *t*-test.

The concordance of distributions that differed from the normal distribution was evaluated using the Mann–Whitney U test. As post hoc testing for group comparisons, Tukey’s HSD test for parametric analysis and Dunn’s test for non-parametric analysis were both used. The average measurements (given as the average of six measurements) for the whole research period, as well as each measurement, were both calculated.

For the comparison analysis of anthropometric measurements gathered at different times, either an ANOVA analysis for repeated measurements or a non-parametric Friedman test was used, depending on how well the distributions matched the normal distribution. The appropriate post hoc testing for intergroup comparisons was Tukey’s HSD test or Friedman’s post-hoc test.

Three-factor analysis of variance (ANOVA) was used to analyze the composite data of anthropometric profiling and segmental changes in the body composition of professional football players in a study about age during the training macrocycle. This allowed us to understand whether there were statistically significant differences between groups concerning various factors and interactions between them. In this particular study, a three-factor ANOVA was used due to three significant factors:

Age group: Football players’ anthropometric values and body composition can change with age due to the body’s growth, maturation, and aging processes. Analysis of age groups identified possible trends or differences in these values across age groups.

Timing (phases of the training macrocycle): There are different phases in the macrocycle of training of professional football players, such as the preparatory period, the competitive period, and the transition period. Anthropometric values and body composition may change, depending on the type of training and intensity in each phase. Time factor analysis determined whether there are changes in the body composition of football players in different phases of the macrocycle.

Limb comparison: Segmental body composition refers to different parts of the body, such as the upper and lower limbs and the body. Differences in body composition between different body parts can be important in terms of a player’s performance, strength, and function. The limb comparison factor allowed us to assess whether there are differences in body composition between different body parts.

A three-factor ANOVA allowed us to analyze the effect of each factor (age group, time, limb comparison) on variables related to anthropometry and the body composition of professional football players.

Statistical tests for small subgroups and multiple comparisons were corrected.

The criterion for statistical significance was *p* < 0.05.

## 3. Results

Athletes in the older age group had significantly higher body height, body mass, BMI, fat-free mass, fat mass, and body fat percentage than younger athletes. Details of the contents of the various body composition parameters are presented in a previous study [16].

Analyzing the segmental body composition according to age showed that the older age group was characterized by higher fat-free mass content in the dominant extremities, assisting extremities, and trunk, of which the details are shown in Table 1.

There were significant differences in the content of fat-free mass among body parts in the age groups in all timepoint measurements, with higher values in the group of athletes aged 26–31 (*p* < 0.05).

Significant variation was found among timepoint measurements 1–6 in the study group. The fat-free mass content in the dominant arm increased between timepoint measurements 1 and 2,6 and 2–5, and decreased between timepoint measurements 2 and 3,4 and 5–6 (*p* < 0.001). In the assisting arm, fat-free mass increased between timepoint measurements 1 and 2,6, and 5–6 (*p* < 0.001). The fat-free mass content in the trunk increased between timepoint measurements 1 and 6 (*p* < 0.001). The fat-free mass content in the dominant leg increased between timepoint measurements 1 and 2,3,4,6 and decreased between timepoint measurements 2,4 and 5 (*p* < 0.0001), while in the assisting leg, it increased between timepoint measurements 1 and 2,3,4,6 and 5–6, and decreased between timepoint measurements 2,3 and 5 (*p* < 0.0001).

The 18–25 age group showed significant variation in fat-free mass content. The fat-free mass content in the dominant arm decreased between timepoint measurements 2 and 5 (*p* < 0.001), in the assisting arm it increased between timepoints 1–2 and 5–6, while it decreased between timepoint measurements 2 and 4,5 (*p* < 0.01). The fat-free mass of the torso decreased between timepoint measurements 2 and 5 (*p* < 0.05). Analyzing changes in lower extremity fat-free mass content showed an increase in the dominant leg between timepoint measurements 5–6 (*p* < 0.01), and in the assisting leg between timepoint measurements 1,5 and 6 (*p* < 0.05).

In the older age group, significant differences in fat-free mass content were found between measurements in the lower extremities. Fat-free mass in the dominant and assisting leg increased between timepoint measurements 1 and 2,3,4,6 and decreased between timepoint measurements 2–5 (*p* < 0.0001; *p* < 0.0001). Detailed information is shown in Figure 1.

Another element is body fat content. Significantly higher body parts fat content was associated with the older age group (*p* < 0.05). In addition, a statistically higher level of visceral fat was shown in this group (*p* < 0.001). Detailed information is presented in Table 2.

There was a significant variation in body fat content between timepoint measurements 1–6 in the study group. Fat mass content in the dominant arm increased between 1,2,3 and 6 timepoint measurements (*p* < 0.01), and fat mass content in the assisting arm increased between 2,3 and 6 timepoint measurements (*p* < 0.01). There was a significant increase in fat content in the trunk and dominant leg between 1,2,3,4 and 6 timepoint measurements (*p* < 0.01; *p* < 0.001), as well as an increase in fat content in the assisting leg between 1,2,3,4 and 6 and 2–5 timepoint measurements (*p* < 0.01). There was a significant increase in visceral fat between 1,2,3 and 5,6 timepoint measurements (*p* < 0.0001). In the younger age group, there was a significant decrease in fat mass between the 1–2 timepoint measurements and an increase in fat mass between the 2–6 timepoint measurements in the dominant arm (*p* < 0.05). Fat mass located in the trunk significantly increased between the 2–6 timepoint measurements (*p* < 0.05). Fat content in the dominant leg increased between the 2,3 and 6 timepoint measurements (*p* < 0.05). No significant changes were indicated in the older age group. The information is shown in Figure 2.

## 4. Discussion

Age is an important determinant of body composition (including, among other things, muscle mass content, fat mass, and visceral fat), related both to individual development and maturity and to an athlete’s training seniority and experience, taking into account, among other things, the number of starts. In elite teams, the age of the athletes varies and is based, among other things, on training ability, endurance, skill, and fitness. Players over 15 can start a professional career with professional clubs, while no age determines the end of a career. The oldest players participating in the elite competition are over 40 years old. A player’s seniority, and therefore, the number of training units and matches they have played, determines the composition of their body mass and the changes that occur [13,15,16].

The purpose of this study was to assess differences in body composition between younger and older professional football players and to evaluate body composition management skills during the training macrocycle. Significant differences were noted between the two elite professional groups in terms of segmental analysis measurements. In addition, confirming the general assumption, differences were shown between the fat-free mass and fat mass content of the athletes’ body components according to the dominant and assisting limb.

In addition, there were no differences in changes in body composition over the macro training cycle in football players by age group. However, changes were found in the overall content of individual components.

The current study was conducted to obtain practical implications. The results obtained will allow training plans to be adapted to changes in body composition at different phases of a football player’s life, ensuring the personalization and optimization of training. They will help understand how age affects body composition and what training interventions are needed to maintain performance. Information about the effects of age and different moments of the macro training cycle on body composition can improve injury prevention strategies. In addition, they can serve as a source of knowledge and data for other studies. They will help tailor training for different developmental stages of athletes and provide a basis for long-term planning of training strategies.

Currently, there are no guidelines defining the benchmark amount of body fat needed to prevent relative energy deficiency in sports (RED-S) in football players during periods occurring in the training macro-cycle. Various values of body composition parameters are described in the research, depending on the level of play, the moment of the season, the position on the field, and age [24,25,26]. The optimal body fat content in football, which helps prevent RED-S, can vary, depending on individual physical characteristics, training intensity, and other factors. Nevertheless, maintaining a healthy and balanced energy balance is key. The key to preventing RED-S in football is a balanced approach to training, nutrition, and recovery that allows athletes to maintain a proper energy balance and achieve an optimal body composition. Age and the specificity of the limb conformation should be taken into account when determining optimal ranges of fat-free mass, fat mass, and body fat percentage values for football players, as the results of the current study indicate. It should be emphasized that these ranges should be determined on a case-by-case basis due to the number of factors that determine optimal composition [26].

The differences in the content of the various parameters are due to the limb preference in activities, such as passing and shots on goal. Footballers perform these activities with the dominant leg, which affects the greater load on the limb and can result in a disproportion in body mass composition. When significant morphological asymmetry is detected in elite athletes, it is essential to monitor the differences and compensate for them through appropriate training and nutrition [13].

The fat-free body content of the upper extremities was similar in the dominant and assisting arms, averaging 4.07 ± 0.42 kg and 4.05 ± 0.42 kg, respectively. Greater discrepancies in the content of fat-free mass were shown in the lower limbs. The average content in the dominant leg was 11.16 ± 1.06 kg, and 11.08 ± 1.01 kg in the assisting leg. Similar differences were not shown in body fat mass in individual limbs. A study by Chao et al. involving 26 Taiwanese athletes showed differences in fat-free mass content between the right and left sides of the lower extremities. The fat-free mass content in the left arm was 2.94 ± 0.44 kg, while in the right arm, it was 2.97 ± 0.41 kg. In contrast, the lower extremities showed differences of 11.42 ± 1.30 kg in the left leg and 11.63 ± 1.24 kg in the right leg; in addition, there were no differences in fat mass between the limbs, similar to the authors’ results [24].

Milsom et al. showed differences in fat-free mass and fat mass between limbs. FFM in the left and right arms was 4.09 ± 0.57 kg and 4.39 ± 0.56 kg, respectively, and in the left and right legs 11.44 ± 1.21 kg and 11.84 ± 1.31 kg, respectively. The fat mass in the left arm averaged 0.5 ± 0.09 kg, and in the right arm averaged 0.56 ± 0.014 kg. In contrast, fat mass in the left leg averaged less than that in the right leg, 1.52 ± 0.37 kg, and 1.62 ± 0.39 kg, respectively [25].

Analyzing the fat-free mass content of the body parts of the athletes in the age groups showed significant differences. Older athletes were characterized by higher fat-free mass content in all body parts (dominant arm *p* < 0.01; assisting arm *p* < 0.01; trunk *p* < 0.01; dominant leg *p* < 0.05; assisting leg *p* < 0.01). A study by Milsom et al. showed a significantly higher proportion of lean body mass in all monitored body segments of players of an English soccer team playing at the highest level of the competition compared to players under 21 and under 18 years of age [27].

An important factor determining the body composition of athletes is the season period. The preparatory period’s goal is to shape the athlete’s fitness and improve his or her performance following a break. Technical and tactical training is extremely intense, with a heavier overall workload than during the competitive phase [28]. The preparatory period is an appropriate time to modify body composition by adjusting nutrition and training loads to the goal the athlete should achieve. Optimal loss of body fat mass while maintaining or increasing fat-free mass is highly relevant to athletic performance. During the preparatory period, there should be dynamic changes in the body mass and composition of the athletes, mainly under the influence of training stimuli and an appropriately applied diet. These changes should primarily occur in the players’ fat and lean mass. The increase in body mass should be mainly caused by an increase in the fat-free mass of the athletes, while its maintenance should be accomplished by a simultaneous reduction in fat mass and increase in FFM. Fat-free mass contributes to increases in the strength and power of athletes [29,30].

The results indicate an increase in the fat-free body mass content of the upper and lower extremities during the preparatory period. The most significant increase in FFM was identified in the dominant and assisting legs from 11.07 ± 1.07 kg to 11.2 ± 1.1 kg and from 11 ± 1.04 kg to 11.12 ± 1.03 kg, respectively. Body fat mass did not change significantly during the preparatory period in any of the analyzed body parts. A study by McEwan et al. analyzed two extremities combined and showed a slight increase in fat-free mass in the upper extremities from 6.8 ± 0.7 kg to 6.84 ± 0.66 kg and an increase in FFM from 22.13 ± 2.07 kg to 22.38 ± 2.08 kg in the lower extremities during the preparatory period. McEwan et al. further found differences in fat mass: fat mass in the lower extremities decreased from 1.02 ± 0.15 kg to 0.91 ± 0.14 kg, and in the lower extremities, from 3.65 ± 0.67 kg to 3.29 ± 0.58 kg; the current study did not show the above changes [31].

After the preparatory period, a lengthier competitive period occurs, which is marked by the occurrence of many match games that require a lot of energy. However, because more recuperation training is performed in between games, the intensity of training is lower than it was during the preceding phase [32]. To maintain an optimal body composition throughout the competition period, the professional athlete needs to adjust his nutrition to the exercise loads happening throughout this time. Athletes who consume more energy than their bodies need may have a rise in body fat, whereas those who consume too little and in the wrong proportions of macronutrients may experience greater muscle catabolism and a loss of mass. Inadequate body fat and muscle mass in elite football players can have a negative impact on performance and health [11,33].

Analyzing segmental body composition showed a decrease in fat-free mass in the dominant arm and the dominant and assisting leg. Evaluating segmental body fat mass, the current study showed an increase in body fat content in the assisting leg and an increase in visceral fat over the starting period. The increase in adipose tissue content, with a concomitant reduction in fat-free mass content during the start period, may have been the result of a reduced training load during the competitive period. During the preparatory period, the training program includes general training and high-intensity conditioning workouts, while a significant number of workouts during the competitive period are related to game tactics, ball possession, and fixed game fragments, which are characterized by a lower load [30].

Analyzing the segmental composition of players’ body mass during the competitive period showed differences between age groups. In the younger age group, over the course of the competitive period, fat-free mass decreased in the dominant and assisting arm and in the trunk area. In contrast, in the older age group, FFM decreased in the dominant and assisting leg.

The training is completely stopped or significantly scaled back during the transition time. Athletes may occasionally participate in recreational sports and unorganized training. The length of the transition phase, the reduction in training units, and the athletes’ level of fitness will vary the kinetics of changes in body composition, which could cause some training-induced adaptations to partially or completely disappear [33]. The incapacity of professional football players to adjust their nutrition to decreased physical activity during the transition phase may result in unfavorable changes in body composition. Other risk factors include limitations on training units and club match participation, sedentary leisure activities, and these risk factors [34]. The increase in body mass resulting from an improper diet affects the athletes’ performance and fitness. Decreased muscle mass caused by a lack of training stimulus can result in decreased strength and endurance, and thus, an increased risk of injury when numerous and intense training units are reintroduced during the preparation period [11], which also showed an increase in lean body mass and fat mass in all body segments analyzed.

Analyzing studies with similar themes, Milanese obtained interesting results. The researchers evaluated seasonal changes in the body composition of 31 players using DXA and anthropometry. The determinations were made three times: before the season, during the season, and after the season. The results of the study indicate a decrease in fat mass and body fat percentage between measurements 1 and 2, and a concomitant increase in lean body mass during this period and between measurements 2 and 3 [35]. However, the test performed by this method, due to potential radiation, should not be performed at high frequency. A greater use for frequent body composition measurements is the BIA method. Another study with a similar theme, but using a different method of measurement, is that conducted by Owen et al. They evaluated body composition changes at five different points during the season in 22 athletes using skinfold measurement. The results indicated a significant decrease in fat mass during the preparation period, similar to our results. No such changes were found at subsequent measurement points. The researchers concluded that significant changes in body composition occur and are common throughout the season [36].

Dual-energy absorptiometry (DEXA) is now the accepted method for determining body composition. However, BIA technology has significantly improved to employ a variety of frequencies and impedance measurements to boost the stability and repeatability of body composition evaluation. When compared to the reference DEXA method, studies among clinical populations, fit athletes, and physically inactive individuals have demonstrated that the DSM-BIA approach used in the InBody 770 is valid and trustworthy [22,23].

This study has some limitations. The study group is not very large; however, it is important to note the fact of the repeatability of measurements: the sport of football is injury-ridden, and often players are excluded from games or training, which was a criterion for exclusion from the study. A strong point of the study is the lack of similar studies in the current scientific literature. Most available studies analyze the content of individual components of body mass in terms of the whole body; if they already focus on segmental analysis, it is only in terms of one or a small number of measurements. In addition, this study can provide practical information for coaches and players, which makes it useful and valuable to the football community.

## 5. Conclusions

Anthropometric profiling and comprehensive body composition analysis are important tools used in preparing athletes for competition. Optimal body composition is an important component of athletic performance and athletic training. Based on research, it is observed that excessive body fat content negatively affects energy expenditure, the power-to-weight ratio, and performance. On the other hand, the optimal content of lean body mass determines the value of individual fitness parameters. Body composition assessment remains one of the best tools available to coaches to monitor body composition for performance and health. Therefore, assessing body composition should be routine practice over the course of a season and multiple playing years.

Changes in the content of lean body mass and fat mass in the age groups were shown; in addition, there was a disproportion in the content of body composition components. Understanding how muscle and fat mass are distributed in the extremities can help coaches adjust training and exercise to improve specific skills. Differences in fat-free mass between limbs can lead to muscle imbalances, which, in turn, can increase the risk of injury. The results of these studies can identify possible imbalances between body sides. This information can be useful for profiling individual players, which can help determine their role on the field and appropriate tactics.

Age is an important determinant of body mass composition, and is related to both individual development and maturity, as well as the player’s seniority and experience. In elite teams, the age of players varies, which significantly affects the discrepancies in the body composition of players and the difficulty of conducting anthropometric profiling. Therefore, the segmented body mass composition assessment allows for a more detailed analysis and customization of training and nutrition plans to meet the individual needs and goals of athletes. This works in favor of their performance, health, and long-term sports development.

Managing body mass composition allows athletes to adapt to changes, such as changes in training intensity, recovery periods, or breaks between seasons. Proper management of body composition is important for an athlete’s long-term development, both in the context of a sports career and general health. All of these aspects combine to emphasize that proper management of body composition by football players is key to achieving their full athletic potential and maintaining good health throughout the season and long term.

## Figures and Tables

**Figure 1 sports-11-00172-f001:**
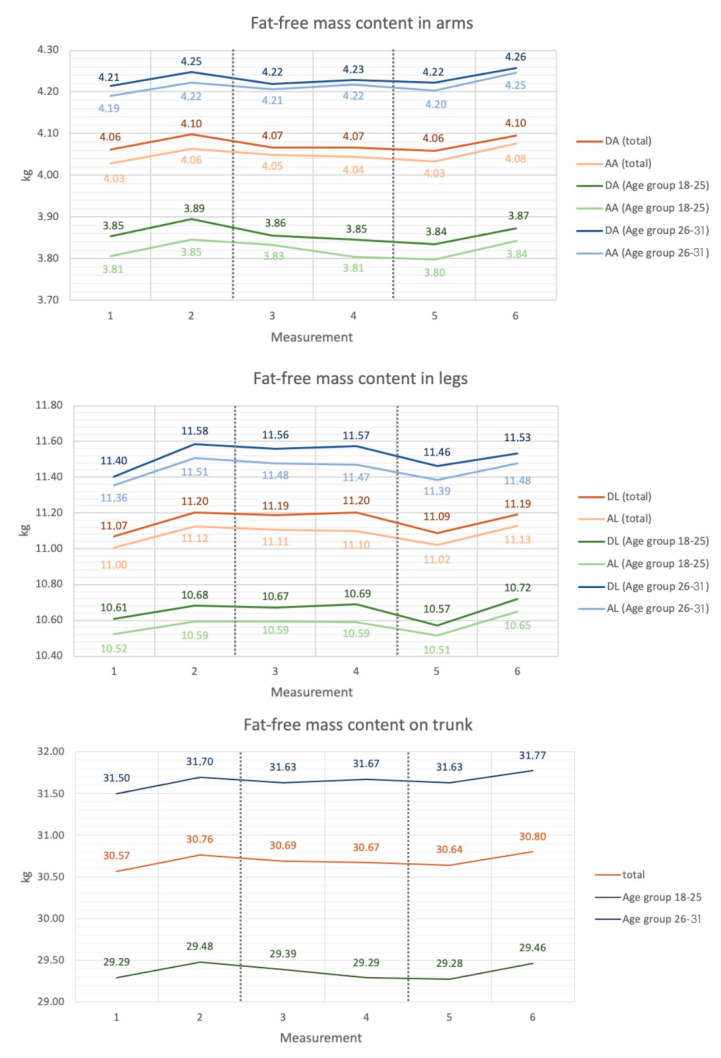
Changes in fat-free mass in body parts of football players in regards to age group (average values): DA-Dominant Arm; AA—Assisting Arm; DL—Dominant Leg; AL—Assisting Leg; Measurement 1–2 = preparatory period, measurement 3–4 = competitive period, measurement 5–6 = transition period.

**Figure 2 sports-11-00172-f002:**
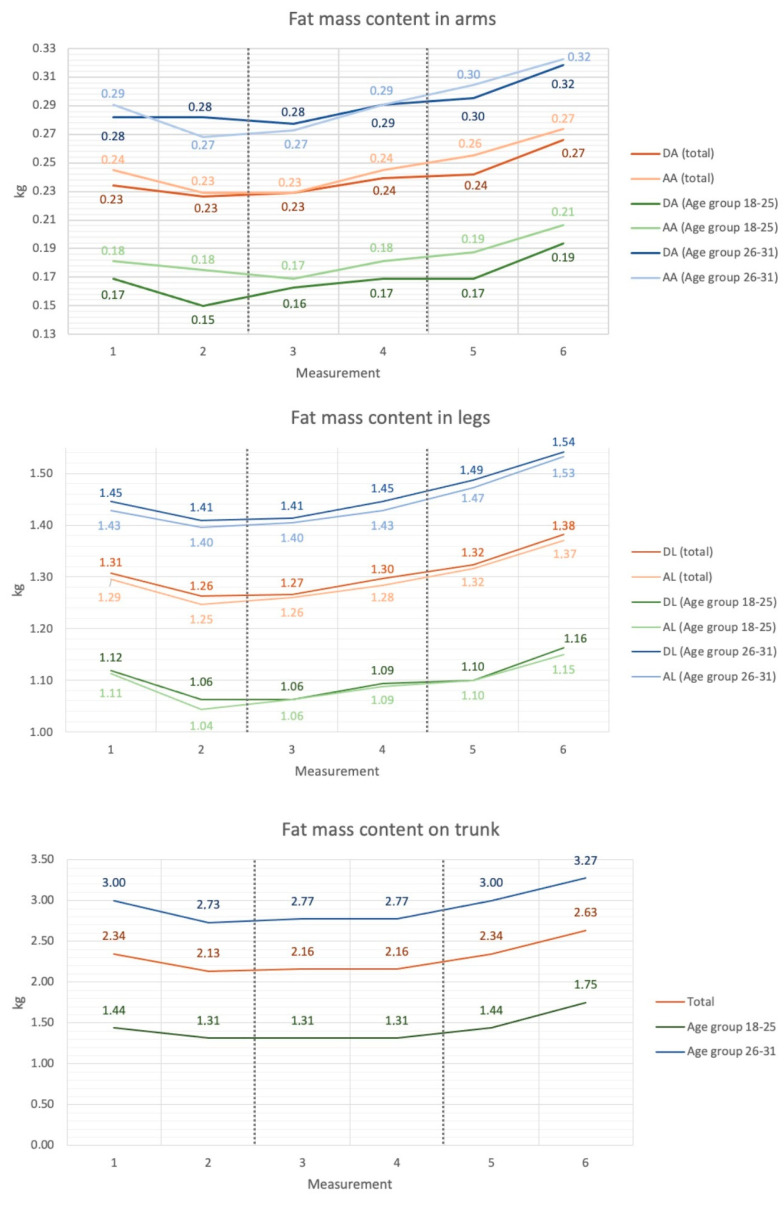
Changes in fat mass in body parts of football players in regards to age group (average values): DA-Dominant Arm; AA—Assisting Arm; DL—Dominant Leg; AL—Assisting Leg; Measurement 1–2 = preparatory period, measurement 3–4 = competitive period, measurement 5–6 = transition period.

**Table 1 sports-11-00172-t001:** Fat-free mass content of the players’ body components (average values).

Variable		Fat-Free Mass [kg]				
		Dominant Arm X ± SD Med (Min–Max)	Assisting Arm X ± SD Med (Min–Max)	Trunk X ± SD Med (Min–Max)	Dominant Leg X ± SD Med (Min–Max)	Assisting Leg X ± SD Med (Min–Max)
Total		4.1 ± 0.4 4.1 (3.8–4.3)	4.1 ± 0.4 4.0 (3.7–4.3)	30.7 ± 2.4 30.5 (29.0–32.4)	11.2 ± 1.1 11.0 (10.3–12.0)	11.1 ± 1.0 11.0 (10.3–11.8)
Age group	18–25	3.9 ± 0.2 ** 3.9 (3.6–4.1)	3.8 ± 0.3** 3.9 (3.6–4.0)	29.4 ± 1.4 ** 29.6 (27.9–30.3)	10.7 ± 0.8 * 10.4 (10.3–11.0)	10.6 ± 0.8 ** 10.4 (10.2–10.9)
	26–31	4.2 ± 0.5 ** 4.2 (4.0–4.5)	4.2 ± 0.5 ** 4.6 (4.0–4.5)	31.67 ± 2.56 ** 31.7 (30.3–33.2)	11.5 ± 1.1 * 11.6 (10.8–12.4)	11.5 ± 1.0 ** 11.5 (10.7–12.2)

X = average; SD = standard deviation; Med = median; Min = minimum; Max = maximum; * = *p* < 0.05; ** = *p* < 0.01.

**Table 2 sports-11-00172-t002:** Fat mass content of the players’ body components (average values).

Variable		Body Fat Mass [kg]					Visceral Fat Level X ± SD Med (Min–Max)
		Dominant Arm X ± SD Med (Min–Max)	Assisting Arm X ± SD Med (Min–Max)	Trunk X ± SD Med (Min–Max)	Dominant Leg X ± SD Med (Min–Max)	Assisting Leg X ± SD Med (Min–Max)	
Total		0.24 ± 0.13 0.19 (0.13–0.38)	0.25 ± 0.13 0.2 (0.15–0.37)	4.07 ± 1.52 3.66 (2.9–5.27)	1.31 ± 0.31 1.23 (1.12–1.53)	1.30 ± 0.31 1.23 (1.12–1.52)	2.29 ± 1.21 2.00 (1.33–3.33)
Age group	18–25	0.17 ± 0.06 * 0.18 (0.1–0.2)	0.18 ± 0.07 ** 0.18 (0.13–0.22)	3.02 ± 0.90 *** 2.92 (2.55–3.54)	1.10 ± 0.21 *** 1.14 (1.03–1.23)	1.09 ± 0.21 *** 1.14 (1.02–1.21)	1.43 ± 0.46 *** 1.33 (1.00–1.75)
	26–31	0.29 ± 0.15 * 0.30 (0.17–0.4)	0.29 ± 0.15 ** 0.28 (0.17–0.4)	4.84 ± 1.42 *** 5.20 (3.63–5.58)	1.46 ± 0.30 *** 1.45 (1.17–1.68)	1.44 ± 0.29 *** 1.43 (1.18–1.65)	2.92 ± 1.21 *** 3.17 (2.00–4.00)

X = average; SD = standard deviation; Med = median; Min = minimum; Max = maximum; * = *p* < 0.05; ** = *p* < 0.01; *** = *p* < 0.001.

## Data Availability

The data presented in this study are available on request from the corresponding author. The data are not publicly available due to privacy.

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
