# Peer review of "Anthropometric Profiling and Changes in Segmental Body Composition of Professional Football Players in Relation to Age over the Training Macrocycle"

_sports, 2023, doi:10.3390/sports11090172_

Round 1

Reviewer 1 Report

Overview

The authors conducted a descriptive study of the change in body composition (FFM and fat mass) over an entire pro soccer season include pre-competitive, competitive, and post-competitive phases.  The authors compared the changes between young (18-25 y) and older (26-31 y) players.  Multiple segmental impedance assessment was used to assess FFM and fat mass.  The authors report that differences were found between the two age groups (older had more FFM and fat mass), differences over time (decreases in FFM and fat mass during the competitive phase and increases afterwards).  It is not clear whether interactive effects were detected but the pattern of change for the age groups and the limbs appears generally the same of both age groups. The authors concluded that age impacts body composition with younger players having less FFM and fat mass (and are leaner, % fat wise) than older players.  Dominant and non-dominant limbs differ with dominant having more FFM and fat mass.  Older players also carry more visceral fat.  Similar changes appear to occur in both groups and both limbs over the macrocycle.

Major Concerns 

Additional information about the methods is needed.

·        * How was impedance converted to FFM and fat mass? What equation was used?

·        * Status of subjects at each measure, specifically efforts to ensure euhydration or documentation of hypo/hyperhydration status, which would affect InBody values.

·        * Statistical methods need more detail as explained below.

Based on what is not reported in the Results, I question whether the authors used the correct statistical analyses.  I would expect a 3-way ANOVA to be used to address the three main effects: age, time, and limbs.  This would lead to multiple interactions to evaluate, and the interactions are of most interest based on the study purpose.  We don’t know the sample size for the two age groups; sample size will impact the ability to detect interactive effects.  If enough attrition occurred and the time-point samples are small, the authors might be justified in using two-way ANOVA with the age groups separated or the limbs separated, but the p level should be adjusted downward to avoid Type 1 errors.  Regardless, all this needs to be added to the Methods or explained, i.e. why the authors took the approach they did.

The Discussion needs revision to remove/delete redundancy with what the reader/reviewer has already been informed of (e.g., lines 262-264, 282-285, etc.).  It should also be reorganized to start with a short paragraph of the purpose and general results, then go into comparisons with the literature and explanations why differences or changes occurred as observed.

Specific Suggestions/Questions

Lines 20-22 “to determine their strengths and weaknesses and to develop successful strength and conditioning programs.  Profiling elite athletes is a useful method for determining skills:” The authors have not addressed these factors.  Better to start with something about the focus of the study - body composition – and how it impacts these factors and potentially the success of soccer players.

Line 34 “…linked to personal growth and maturity…:” Did the authors assess growth and biological maturity in this student? If not, this statement cannot be made.

Line 85: The authors state that a purpose of the study was to “manage” body composition over the macrocycle.  If this is true, it implies some intervention was made and that would mean this is not a descriptive study.  Is that the case?  This needs clarification.

Lines 131-135: State how many subjects were included in each age group. 

Lines 135-136: What do the authors mean that exclusion and inclusion criteria were considered?  The reviewer assumes that determine which subjects’ data were included for hypothesis testing.  What were the criteria?

Lines 146-153: This section describes how impedance was measured but there isn’t a description of how impedance was converted to fat-free mass.  What equation was used?

Lines 163-167:  What efforts were made to ensure subjects were euhydrated?  Were any tests, e.g., urine specific gravity, made to establish subjects were well hydrated?  BIA methods are affected by changes in body water.  For the duration of this study, there are sure to be seasonal changes that alter the environmental (weather) and pose unequal heat stress that might alter general hydration status and body mass due to sweating and inadequate fluid replacement.  If acute hydration status was not accounted for, that should be added to the Discussion as a limitation of the study.

Lines 168-169: If the authors are attempting to say that DEXA is the gold standard, that is not the case.  A multi-component approach that combines densitometry, body water (tracer method), and bone mineral (DEXA) measurements is the gold standard.  However, as a single measurement, the DEXA would approximate the multi-component criterion.

Lines 192-196: Additional information is needed on the primary design for statistical analysis.  With age, time, and limb (dominant vs assisting), the authors must have used a three-way ANOVA, correct? If so, the most important outcomes would be the interactions, e.g., how one limb compares to another differently in one age group over time.  None of this is reported in the results.   

Lines 201-202: Was the younger group leaner (lower body fat %), statistically?

Table 1

·        Keep the number of decimals presented consistent within a variable.  Authors could present just 1 decimal place for all variables.  Likewise for Table 2 and 3.

·        Define X and SD in the footnote.  Also, keep abbreviations consistent between tables. Table 2 uses S not SD. Again, likewise for Table 3.

Figure 1 and Figure 2 use ordinal levels of measurement for the 6 test dates.  However, according to Lines 116-121,  unequal intervals exist between measurements – some less than a month (Jan to Feb), some almost two months apart (March to April/May).   Figure 1 and 2 should be redone to reflect this.  In the form, both figures will be more telling of the true patterns of change.

It would also be helpful on these figures to indicate the three phases of the study - preparatory period, the competitive period, and the post competitive period.  Consider using shading or dotted vertical lines on the graph.

Line 262-264:  This should be part of the Methods or at the beginning of the Results. Or be deleted since the authors described this already.  Likewise, lines 282-285 should be deleted since we’ve already been told this.

Line 266: What is the “optimal body composition” of pro soccer players?  Is there a certain amount of fat (stored energy) needed to prevent relative energy deficiency in sport (RED-S) in soccer players during the macrocycle?  If one does not exist the authors should state so and use that as justification for the existence of this study in the literature, and that age and possibly limb specificity need to be considered when establish an optimal value or range of values for FFM, fat mass (and precent body fat) for soccer players.

Line 328 Typo – “od-be.”

Lines 380-386: This should be the first paragraph of the Discussion – a summary of the main findings would be helpful at the start.  Following that, the authors can proceed with explaining why changes and differences occurred as observed.

Line 398 “correct body composition:” Is there a correct body composition for soccer?  No standard or range for ideal was provided by the authors to see how their subjects compared.

Ines 399-400 “Based on research, it is observed that body fat content affects energy expenditure, power-to-weight ratio, and performance:”  The authors previously used a similar vague statement.  In the next statement, the authors allude to the value of the lean body mass in performance.  Presumably, they mean LBM promotes or enhances performance and fat mass detracts.  If so, this is an overstatement and, if read by coaches or athletes, may set them on the road to trying to lower body fat when it is not warranted.

Line 483: Ref 29 1st author “McEwa” has name misspelled.

Related, there are other similar papers in the literature.  Since this is somewhat of a defense of using the InBody technology, how did the results of the current study compare with those below who used different methods?

·        Using DEXA: Milanese C, Cavedon V, Corradini G, De Vita F, Zancanaro C. Seasonal DXA-measured body composition changes in professional male soccer players.  J Sports Sci. 2015;33(12):1219-28. doi: 10.1080/02640414.2015.1022573. Epub 2015 Mar 16.

·        Using skinfolds: Owen AL, Lago-Peñas C, Dunlop G, Mehdi R, Chtara M, Dellal A. Seasonal Body Composition Variation Amongst Elite European Professional Soccer Players: An Approach of Talent Identification. J Hum Kinet. 2018 Jun 13;62:177-184. doi: 10.1515/hukin-2017-0132. eCollection 2018 Jun.

No specific suggestions.

Author Response

Response to Reviewer 1

Thank you so much for taking the time to evaluate our work. We have tried to incorporate all your valuable suggestions. We have incorporated all the suggestions in the text. Thank you very much. If we could improve our work in any way, please let us know.

Comment 1. How was impedance converted to FFM and fat mass? What equation was used?

The study used a standardized BIA InBody 770 body composition analyzer. The analyzer has specialized LookinBody software that calculates by substituting parameters into appropriate formulas.

Comment 2. Status of subjects at each measure, specifically efforts to ensure euhydration or documentation of hypo/hyperhydration status, which would affect InBody values.

The extracellular water index was determined during each measurement, it is the ratio of extracellular water to total water content and is an important indicator of water balance in the body. This parameter indicates possible edema and water retention in the body. In addition, the athletes were included in a standard hydration protocol after physical activity.

Altarriba-Bartes, A., Pena, J., Vicens-Bordas, J., Mila-Villaroel, R., & Calleja-Gonzalez, J. (2020). Post-competition recovery strategies in elite male soccer players. Effects on performance: A systematic review and meta-analysis. PLoS One, 15(10), e0240135.

Comment 3. Statistical methods need more detail as explained below.

Based on what is not reported in the Results, I question whether the authors used the correct statistical analyses.  I would expect a 3-way ANOVA to be used to address the three main effects: age, time, and limbs.  This would lead to multiple interactions to evaluate, and the interactions are of most interest based on the study purpose.  We don’t know the sample size for the two age groups; sample size will impact the ability to detect interactive effects.  If enough attrition occurred and the time-point samples are small, the authors might be justified in using two-way ANOVA with the age groups separated or the limbs separated, but the p level should be adjusted downward to avoid Type 1 errors.  Regardless, all this needs to be added to the Methods or explained, i.e. why the authors took the approach they did.

In this publication, the purpose of the study was to describe and compare the body composition and anthropometric profiles of players and the management of body composition throughout the round in the 2020-2021 season. The comparison was made on the basis of the division of the studied group of footballers in relation to their age, as described above, why such two age groups were adopted. As for the size of the study group, it is due to the number of professional football players playing in the study teams. Due to the reasons for the very specific study group, and the criteria used for the inclusion and exclusion of football players from the study, the size of the study group is 58.

In our statistical analysis, we used a one-way ANOVA analysis of variance. In our study, we used the ANOVA test to analyze repeated measurements, as described in detail in the methodology, including specific post-hoc tests for evaluating multiple comparisons of measurements. Analysis of variance alone allows us to assess whether or not differences exist in the compared averages. On the other hand, the post-hoc tests used subsequently were used for group comparisons.

The significance of the differences between the means in the age groups (19-25 years, 26-31 years) was assessed using the Student's t-test. Tukey's HSD test for parametric analysis and Dunn's test for nonparametric analysis were used as post-hoc tests for group comparisons.

For comparative analysis of anthropometric measurements collected at different times, ANOVA analysis for repeated measures or Friedman's non-parametric test was used, depending on how well the distributions fit a normal distribution. The appropriate post-hoc test for intergroup comparisons was Tukey's HSD test or Friedman's post-hoc test.

Statistical tests for small subgroups and multiple comparisons were adjusted, due to the distribution of the study group. We adopted the criterion of statistical significance of p < 0.05, which is a standard criterion in this type of study.

We did not use multivariate ANOVA analysis in our study therefore they are not described in the statistical methods.

Comment 4. The Discussion needs revision to remove/delete redundancy with what the reader/reviewer has already been informed of (e.g., lines 262-264, 282-285, etc.).  It should also be reorganized to start with a short paragraph of the purpose and general results, then go into comparisons with the literature and explanations why differences or changes occurred as observed.

Corrected according to the reviewer's comments. Thank you very much for your valuable suggestions, which definitely improve the readability of the survey.

Comment 5. Lines 20-22 “to determine their strengths and weaknesses and to develop successful strength and conditioning programs.  Profiling elite athletes is a useful method for determining skills:” The authors have not addressed these factors.  Better to start with something about the focus of the study - body composition – and how it impacts these factors and potentially the success of soccer players.

Corrected according to the reviewer's comments.

Comment 6. Line 34 “…linked to personal growth and maturity…:” Did the authors assess growth and biological maturity in this student? If not, this statement cannot be made.

Corrected according to the reviewer's comments.

Comment 7. Line 85: The authors state that a purpose of the study was to “manage” body composition over the macrocycle.  If this is true, it implies some intervention was made and that would mean this is not a descriptive study.  Is that the case?  This needs clarification.

According to Griffin's definition, management is a set of activities (planning, organizing, motivating, controlling) directed at an organization's resources (human, financial, physical, information) used with the intention of achieving the organization's goals. The authors, through the use of this term, aim to determine whether players are able to achieve the changes in body composition desired for each period.

Comment 8. Lines 131-135: State how many subjects were included in each age group.

Thank you very much for all your comments, all have been corrected.

Comment 9. Lines 135-136: What do the authors mean that exclusion and inclusion criteria were considered?  The reviewer assumes that determine which subjects’ data were included for hypothesis testing.  What were the criteria?

The inclusion and exclusion criteria are described in detail in lines 97-105.

Comment 10. Lines 146-153: This section describes how impedance was measured but there isn’t a description of how impedance was converted to fat-free mass.  What equation was used?

The study used a standardized BIA InBody 770 body composition analyzer. The analyzer has specialized LookinBody software that calculates body composition parameters by substituting them into appropriate formulas. All information about the device is available at the link:

https://inbodyusa.com/studies/

Comment 11. Lines 163-167:  What efforts were made to ensure subjects were euhydrated?  Were any tests, e.g., urine specific gravity, made to establish subjects were well hydrated?  BIA methods are affected by changes in body water.  For the duration of this study, there are sure to be seasonal changes that alter the environmental (weather) and pose unequal heat stress that might alter general hydration status and body mass due to sweating and inadequate fluid replacement.  If acute hydration status was not accounted for, that should be added to the Discussion as a limitation of the study.

The extracellular water index was determined during each measurement, it is the ratio of extracellular water to total water content and is an important indicator of water balance in the body. This parameter indicates possible edema and water retention in the body. In addition, the athletes were included in a standard hydration protocol after physical activity.

Altarriba-Bartes, A., Pena, J., Vicens-Bordas, J., Mila-Villaroel, R., & Calleja-Gonzalez, J. (2020). Post-competition recovery strategies in elite male soccer players. Effects on performance: A systematic review and meta-analysis. PLoS One, 15(10), e0240135.

Comment 12. Lines 168-169: If the authors are attempting to say that DEXA is the gold standard, that is not the case.  A multi-component approach that combines densitometry, body water (tracer method), and bone mineral (DEXA) measurements is the gold standard.  However, as a single measurement, the DEXA would approximate the multi-component criterion.

The authors did not use the phrase "gold standard" however, they also described this method of body composition assessment in this section due to its widespread use in the sport of football. We are aware of other methods of assessment however in population studies these two methods are the most popular and the superiority of DEXA over BIA is often described. The excerpt is intended to justify the chosen method.

Comment 13. Lines 192-196: Additional information is needed on the primary design for statistical analysis.  With age, time, and limb (dominant vs assisting), the authors must have used a three-way ANOVA, correct? If so, the most important outcomes would be the interactions, e.g., how one limb compares to another differently in one age group over time.  None of this is reported in the results.  

In our statistical analysis, we used a one-way ANOVA analysis of variance. In our study, we used the ANOVA test to analyze repeated measurements, as described in detail in the methodology, including specific post-hoc tests for evaluating multiple comparisons of measurements. Analysis of variance alone allows us to assess whether or not differences exist in the compared averages. On the other hand, the post-hoc tests used subsequently were used for group comparisons.

The significance of the differences between the means in the age groups (19-25 years, 26-31 years) was assessed using the Student's t-test. Tukey's HSD test for parametric analysis and Dunn's test for nonparametric analysis were used as post-hoc tests for group comparisons.

For comparative analysis of anthropometric measurements collected at different times, ANOVA analysis for repeated measures or Friedman's non-parametric test was used, depending on how well the distributions fit a normal distribution. The appropriate post-hoc test for intergroup comparisons was Tukey's HSD test or Friedman's post-hoc test.

We did not use multivariate ANOVA analysis in our study therefore they are not described in the statistical methods.

Comment 14. Lines 201-202: Was the younger group leaner (lower body fat %), statistically?

The younger group was statistically leaner, with a lower percentage body fat. Added relevant information in the text.

Comment 15. Table 1. Keep the number of decimals presented consistent within a variable.  Authors could present just 1 decimal place for all variables.  Likewise for Table 2 and 3. Define X and SD in the footnote.  Also, keep abbreviations consistent between tables. Table 2 uses S not SD. Again, likewise for Table 3.

Corrected according to the reviewer's comments.

Comment 16. Figure 1 and Figure 2 use ordinal levels of measurement for the 6 test dates.  However, according to Lines 116-121,  unequal intervals exist between measurements – some less than a month (Jan to Feb), some almost two months apart (March to April/May).   Figure 1 and 2 should be redone to reflect this.  In the form, both figures will be more telling of the true patterns of change.

The exact dates of the measurements are described in the survey methodology. The horizontal axis of the graph represents measurement numbers and does not reflect the actual passage of time.

Comment 16. It would also be helpful on these figures to indicate the three phases of the study - preparatory period, the competitive period, and the post competitive period.  Consider using shading or dotted vertical lines on the graph.

Corrected according to the reviewer's comments.

Comment 17. Line 262-264:  This should be part of the Methods or at the beginning of the Results. Or be deleted since the authors described this already.  Likewise, lines 282-285 should be deleted since we’ve already been told this.

Corrected according to the reviewer's comments.

Comment 18. Line 266: What is the “optimal body composition” of pro soccer players?  Is there a certain amount of fat (stored energy) needed to prevent relative energy deficiency in sport (RED-S) in soccer players during the macrocycle?  If one does not exist the authors should state so and use that as justification for the existence of this study in the literature, and that age and possibly limb specificity need to be considered when establish an optimal value or range of values for FFM, fat mass (and precent body fat) for soccer players.

Thank you very much for your valuable tips. I have expanded the discussion piece to include the aspect suggested by the reviewer.

Comment 19. Line 328 Typo – “od-be.”

Corrected according to the reviewer's comments.

Comment 20. Lines 380-386: This should be the first paragraph of the Discussion – a summary of the main findings would be helpful at the start.  Following that, the authors can proceed with explaining why changes and differences occurred as observed.

Corrected according to the reviewer's comments.

Comment 21. Line 398 “correct body composition:” Is there a correct body composition for soccer? No standard or range for ideal was provided by the authors to see how their subjects compared.

Changed in the text as suggested by the reviewer. Optimal body composition seems to be a more appropriate statement. Different values are described in studies and there are no clear standards.

Comment 22. Ines 399-400 “Based on research, it is observed that body fat content affects energy expenditure, power-to-weight ratio, and performance:”  The authors previously used a similar vague statement.  In the next statement, the authors allude to the value of the lean body mass in performance.  Presumably, they mean LBM promotes or enhances performance and fat mass detracts.  If so, this is an overstatement and, if read by coaches or athletes, may set them on the road to trying to lower body fat when it is not warranted.

Corrected according to the reviewer's comments.

Comment 23. Line 483: Ref 29 1st author “McEwa” has name misspelled.

Corrected according to the reviewer's comments. I am very sorry for the typing error.

Comment 24. Related, there are other similar papers in the literature.  Since this is somewhat of a defense of using the InBody technology, how did the results of the current study compare with those below who used different methods?

  • Using DEXA: Milanese C, Cavedon V, Corradini G, De Vita F, Zancanaro C. Seasonal DXA-measured body composition changes in professional male soccer players. J Sports Sci. 2015;33(12):1219-28. doi: 10.1080/02640414.2015.1022573. Epub 2015 Mar 16.

  • Using skinfolds: Owen AL, Lago-Peñas C, Dunlop G, Mehdi R, Chtara M, Dellal A. Seasonal Body Composition Variation Amongst Elite European Professional Soccer Players: An Approach of Talent Identification. J Hum Kinet. 2018 Jun 13;62:177-184. doi: 10.1515/hukin-2017-0132. eCollection 2018 Jun.

Thank you very much for pointing out the research. Of course, the relevant section of the text in the discussion section based on them has been added.

Thank you for your help. Your guidance is invaluable.

Kind regards,

Authors.

Reviewer 2 Report

This paper analyses the anthropometric profile of male professional football players in two Polish league teams. It is a continuation of previous work (Front Nutrients 2022 Nov 29;9:981894. doi: 10.3389/fnut.2022.981894 and Nutrients 2023 Jan 30;15(3):705. doi: 10.3390/nu15030705) that both focused much more on whole body composition and nutritional status, with the differences between young and older athletes measured during various timepoints in the macrocycle.

The current paper goes into detail of the limb composition of the athletes. The paper should more specifically mention this focus in the abstract, the introduction and the results section, and introduce better that the current segmental analysis is a continuation of previously executed work and that data have been published elsewhere.

The discussion needs to be rewritten starting with main results from the current data presentation (lines 282-301, a section that puts these results into comparison with previous other work, why any of these profiling anthropometric data are relevant to training and performance (as already present). But omitted or densed-down can be the current 2nd and 3rd paragraphs. The first paragraph of the discussion is redundant.

Please address the following issues in your re-submmitted paper:

Abstract

Lines 20/21 should read: ‘Body composition is an important indicator of an athlete's overall health and physical fitness, and thus anthropometric profiling of elite athletes is a useful method for determining their skills, strengths and weaknesses to develop successful strength and conditioning programs.’

Lines 23/26 should read: ‘The purpose of this study was to describe and compare the body composition and anthropometric profiles of players using the Direct Segmental Multi-Frequency Bio-Electrical Impedance Analysis method, and to manage body composition throughout the round in the 2020-2021 season.’

Lines 26/27 should read: ‘The investigation was carried out during the Polish football league PKO BP Ekstraklasa spring round of the football season 2020–2021 in which male football players participated.’ Here the authors need to indicate the different groups! Age groups: 18 - 25 (n=?) versus ages 26 – 31 (n=?).

Line 28 should read: ‘ At various stages during the macrocycle, the subjects underwent six different body composition analyses.’

Lines 29-34: be more specific about the groups. Are you comparing a young to a older group of players? Measures 1-6, should be replaced by ‘timepoints measurements 1-6’.

Lines 33-36. Formulate conclusion according to results, not these general statements.

Introduction

Line 57 should read: ‘One measure to assess performance progress is body composition assessment.’

Line 64 should read ‘In one round, three periods are distinguished: …’

Line 70/71 should read ‘or significant reduction in training participation.’

Methods:

Lines 93/96: should read ‘This manuscript is a continuation of previous work of which the used methodology has been described previously [36,37].’

Lines 79/100 should read ‘The inclusion criterion for participation in the study were: to be a professional football player in one of the clubs involved (player having signed a professional contract), agreed to participate in the study, and not have any long-term injuries that would prevent them from participating in training and games for the duration of the seven months study.

Line 101/105: should read ‘Exclusion criteria were [36,37]: transferred from one team to another during the study period, absence from practice and competition lasting at least 14 days due to illness, injury, isolation, or quarantine as a result of the COVID-19 pandemic, and failure to participate in at least 1 of the 6 measurements for any other reason than those mentioned above.’

Line 115 should read: ‘Measurements were made at the following macrocycle timepoints’

Section 2.2 should mention the number of participants per age group. Preferably in line 131/132 (n=?)

Lines 168/174 should be used in the discussion, not in the methods.

Results

Tables 1,2,3 ‘Age Group’, ‘18-25’ and ‘26-31’ should be positioned horizontally.

Table 3 is missing its title!

Table 1 results are identical numbers as Table 1 of Nutrients 2023, 15, 705. https://doi.org/10.3390/nu15030705. Hence, should be omitted. References to the previous work should be given here, but not the repetition of Height, BM, BMI, FFM, FM in average + SD, median and range. The data have been given in Nutrients 2023, 15, 705.

Lines 205/207 should read ‘Analyzing the segmental body composition according to age, showed that the older age group was characterized by higher fat-free mass content in the dominant extremities, assisting extremities, and trunk, of which the details are shown in Table 2.’

Lines 210-211 should read ‘There were statistically significant differences in the content of fat-free mass in body parts among age groups in all timepoint measurements, with higher values in the group of athletes aged 26-31 (p<0.05).’ I would reposition this text, directly following lines 205-207, not having the Table separating the two blocks of text.

Lines 242-254: Please replace ‘statistically significant’ with ‘significant’ and replace ‘measurements’ with ‘timepoint measurements’, and indicate this text is referring to either Figure 1 or 2.

Discussion

Omit ‘author’s study’ and ‘authors’ results’, and replace with ‘previous analysis’ or similar.

Discussion needs to be rewritten starting with main results from the current data presentation (lines 282-301, a section that puts these results into comparison with previous other work, why any of these profiling anthropometric data are relevant to training and performance (as already present). But omitted or densed-down can be the current 2nd and 3rd paragraphs. The first paragraph of the discussion is redundant.

Please address the following issues in your re-submmitted paper:

Abstract

Lines 20/21 should read: ‘Body composition is an important indicator of an athlete's overall health and physical fitness, and thus anthropometric profiling of elite athletes is a useful method for determining their skills, strengths and weaknesses to develop successful strength and conditioning programs.’

Lines 23/26 should read: ‘The purpose of this study was to describe and compare the body composition and anthropometric profiles of players using the Direct Segmental Multi-Frequency Bio-Electrical Impedance Analysis method, and to manage body composition throughout the round in the 2020-2021 season.’

Lines 26/27 should read: ‘The investigation was carried out during the Polish football league PKO BP Ekstraklasa spring round of the football season 2020–2021 in which male football players participated.’ Here the authors need to indicate the different groups! Age groups: 18 - 25 (n=?) versus ages 26 – 31 (n=?).

Line 28 should read: ‘ At various stages during the macrocycle, the subjects underwent six different body composition analyses.’

Lines 29-34: be more specific about the groups. Are you comparing a young to a older group of players? Measures 1-6, should be replaced by ‘timepoints measurements 1-6’.

Lines 33-36. Formulate conclusion according to results, not these general statements.

Introduction

Line 57 should read: ‘One measure to assess performance progress is body composition assessment.’

Line 64 should read ‘In one round, three periods are distinguished: …’

Line 70/71 should read ‘or significant reduction in training participation.’

Methods:

Lines 93/96: should read ‘This manuscript is a continuation of previous work of which the used methodology has been described previously [36,37].’

Lines 79/100 should read ‘The inclusion criterion for participation in the study were: to be a professional football player in one of the clubs involved (player having signed a professional contract), agreed to participate in the study, and not have any long-term injuries that would prevent them from participating in training and games for the duration of the seven months study.

Line 101/105: should read ‘Exclusion criteria were [36,37]: transferred from one team to another during the study period, absence from practice and competition lasting at least 14 days due to illness, injury, isolation, or quarantine as a result of the COVID-19 pandemic, and failure to participate in at least 1 of the 6 measurements for any other reason than those mentioned above.’

Line 115 should read: ‘Measurements were made at the following macrocycle timepoints’

Section 2.2 should mention the number of participants per age group. Preferably in line 131/132 (n=?)

Lines 168/174 should be used in the discussion, not in the methods.

Results

Tables 1,2,3 ‘Age Group’, ‘18-25’ and ‘26-31’ should be positioned horizontally.

Table 3 is missing its title!

Table 1 results are identical numbers as Table 1 of Nutrients 2023, 15, 705. https://doi.org/10.3390/nu15030705. Hence, should be omitted. References to the previous work should be given here, but not the repetition of Height, BM, BMI, FFM, FM in average + SD, median and range. The data have been given in Nutrients 2023, 15, 705.

Lines 205/207 should read ‘Analyzing the segmental body composition according to age, showed that the older age group was characterized by higher fat-free mass content in the dominant extremities, assisting extremities, and trunk, of which the details are shown in Table 2.’

Lines 210-211 should read ‘There were statistically significant differences in the content of fat-free mass in body parts among age groups in all timepoint measurements, with higher values in the group of athletes aged 26-31 (p<0.05).’ I would reposition this text, directly following lines 205-207, not having the Table separating the two blocks of text.

Lines 242-254: Please replace ‘statistically significant’ with ‘significant’ and replace ‘measurements’ with ‘timepoint measurements’, and indicate this text is referring to either Figure 1 or 2.

Discussion

Omit ‘author’s study’ and ‘authors’ results’, and replace with ‘previous analysis’ or similar.

Discussion needs to be rewritten starting with main results from the current data presentation (lines 282-301, a section that puts these results into comparison with previous other work, why any of these profiling anthropometric data are relevant to training and performance (as already present). But omitted or densed-down can be the current 2nd and 3rd paragraphs. The first paragraph of the discussion is redundant.

Author Response

Response to Reviewer 2

Thank you so much for taking the time to evaluate our work. We have tried to incorporate all your valuable suggestions. We have incorporated all the suggestions in the text. Thank you very much. If we could improve our work in any way, please let us know.

Comment 1. The current paper goes into detail of the limb composition of the athletes. The paper should more specifically mention this focus in the abstract, the introduction and the results section, and introduce better that the current segmental analysis is a continuation of previously executed work and that data have been published elsewhere.

Corrected according to the reviewer's comments. Thank you very much for your valuable suggestions, which definitely improve the readability of the survey.

Comment 2. The discussion needs to be rewritten starting with main results from the current data presentation (lines 282-301, a section that puts these results into comparison with previous other work, why any of these profiling anthropometric data are relevant to training and performance (as already present). But omitted or densed-down can be the current 2nd and 3rd paragraphs. The first paragraph of the discussion is redundant.

Corrected according to the reviewer's comments.

Comment 3. Lines 20/21 should read: ‘Body composition is an important indicator of an athlete's overall health and physical fitness, and thus anthropometric profiling of elite athletes is a useful method for determining their skills, strengths and weaknesses to develop successful strength and conditioning programs.’

Corrected according to the reviewer's comments.

Comment 4. Lines 23/26 should read: ‘The purpose of this study was to describe and compare the body composition and anthropometric profiles of players using the Direct Segmental Multi-Frequency Bio-Electrical Impedance Analysis method, and to manage body composition throughout the round in the 2020-2021 season.’

Corrected according to the reviewer's comments. Thank you very much for your valuable suggestions, which definitely improve the readability of the survey.

Comment 5. Lines 26/27 should read: ‘The investigation was carried out during the Polish football league PKO BP Ekstraklasa spring round of the football season 2020–2021 in which male football players participated.’ Here the authors need to indicate the different groups! Age groups: 18 - 25 (n=?) versus ages 26 – 31 (n=?).

Corrected according to the reviewer's comments.

Comment 6. Line 28 should read: ‘ At various stages during the macrocycle, the subjects underwent six different body composition analyses.’

Corrected according to the reviewer's comments.

Comment 7. Lines 29-34: be more specific about the groups. Are you comparing a young to a older group of players? Measures 1-6, should be replaced by ‘timepoints measurements 1-6’.

Thank you very much for all your comments, all have been corrected.

Comment 8. Lines 33-36. Formulate conclusion according to results, not these general statements.

Thank you very much for all your comments, all have been corrected.

Comment 9. Line 57 should read: ‘One measure to assess performance progress is body composition assessment.’

Corrected according to the reviewer's comments.

Comment 10. Line 64 should read ‘In one round, three periods are distinguished: …’

Corrected according to the reviewer's comments.

Comment 11. Line 70/71 should read ‘or significant reduction in training participation.’

Corrected according to the reviewer's comments.

Comment 12. Lines 93/96: should read ‘This manuscript is a continuation of previous work of which the used methodology has been described previously [36,37].’

Corrected according to the reviewer's comments.

Comment 13. Lines 79/100 should read ‘The inclusion criterion for participation in the study were: to be a professional football player in one of the clubs involved (player having signed a professional contract), agreed to participate in the study, and not have any long-term injuries that would prevent them from participating in training and games for the duration of the seven months study.

Corrected according to the reviewer's comments.

Comment 14. Line 101/105: should read ‘Exclusion criteria were [36,37]: transferred from one team to another during the study period, absence from practice and competition lasting at least 14 days due to illness, injury, isolation, or quarantine as a result of the COVID-19 pandemic, and failure to participate in at least 1 of the 6 measurements for any other reason than those mentioned above.’

Corrected according to the reviewer's comments.

Comment 15. Line 115 should read: ‘Measurements were made at the following macrocycle timepoints’

Corrected according to the reviewer's comments.

Comment 16. Section 2.2 should mention the number of participants per age group. Preferably in line 131/132 (n=?)

Thank you very much for all your comments, all have been corrected.

Comment 16. Lines 168/174 should be used in the discussion, not in the methods.

Corrected according to the reviewer's comments.

Comment 17. Tables 1,2,3 ‘Age Group’, ‘18-25’ and ‘26-31’ should be positioned horizontally.

Corrected according to the reviewer's comments.

Comment 18. Table 3 is missing its title!

I am very sorry for the oversight. The table title has been added.

Comment 19. Table 1 results are identical numbers as Table 1 of Nutrients 2023, 15, 705. https://doi.org/10.3390/nu15030705. Hence, should be omitted. References to the previous work should be given here, but not the repetition of Height, BM, BMI, FFM, FM in average + SD, median and range. The data have been given in Nutrients 2023, 15, 705.

Thank you very much for your suggestion. We have removed the Tables from the text.

Comment 20. Lines 205/207 should read ‘Analyzing the segmental body composition according to age, showed that the older age group was characterized by higher fat-free mass content in the dominant extremities, assisting extremities, and trunk, of which the details are shown in Table 2.’

Corrected according to the reviewer's comments.

Comment 21. Lines 210-211 should read ‘There were statistically significant differences in the content of fat-free mass in body parts among age groups in all timepoint measurements, with higher values in the group of athletes aged 26-31 (p<0.05).’ I would reposition this text, directly following lines 205-207, not having the Table separating the two blocks of text.

Corrected according to the reviewer's comments.

Comment 22. Lines 242-254: Please replace ‘statistically significant’ with ‘significant’ and replace ‘measurements’ with ‘timepoint measurements’, and indicate this text is referring to either Figure 1 or 2.

Corrected according to the reviewer's comments.

Comment 23. Omit ‘author’s study’ and ‘authors’ results’, and replace with ‘previous analysis’ or similar.

Corrected according to the reviewer's comments.

Comment 24. Discussion needs to be rewritten starting with main results from the current data presentation (lines 282-301, a section that puts these results into comparison with previous other work, why any of these profiling anthropometric data are relevant to training and performance (as already present). But omitted or densed-down can be the current 2nd and 3rd paragraphs. The first paragraph of the discussion is redundant.

Thank you very much for your valuable suggestions. Of course, the discussion has been edited.

Thank you for your help. Your guidance is invaluable.

Kind regards,

Authors.

Round 2

Reviewer 1 Report

Overview

 The authors have addressed some of my concerns.  I still have a number of general (major) concerns and many specific suggestions.

 Major Concerns

 The abstract is improved but still needs much work.  For example, nothing is stated about the body composition importance in football.  It isn’t until the 10th line of the abstract that football is mentioned.

The authors did not address my prior concern re what type of ANOVA was used such as whether a 3-way ANOVA was used since they have three factors, age group by time by limb comparisons.  If not 3-way, why?  The description and/or justification is needed.

“Optimal body composition” continues to be expressed but there is no indication of what optimal is for football.  Isn’t the role of this study to help establish what optimal is or the range for optimal? And what should be assessed (limbs)?

Specific Suggestions/Questions

Lines 21-22 “…anthropometric profiling of elite athletes is a useful method for determining their skills, strengths, and weaknesses…:” Does this profiling truly tell us the skills etc. of the athlete? Or does it help us track underlining factors that contribute to performance?

Line 23 “…one of the finest tools available to coaches…:” Remove the superlative “finest.”  Keep your objectivity.  And to be honest, I don’t know that many sport coaches that use body composition instruments.  At least in the US.  Maybe the strength and conditioning coaches.

Lines 23-24 “…to track body composition for performance and health is the body composition assessment:”  The redundancy of body composition makes this nonsensical.  How about “…to track correlates of performance and health is routine body composition assessment?”

Lines 24-26 “According to a study, an excessive amount of body fat significantly impacts performance, power-to-weight ratio, and energy expenditure. However, the importance of individual fitness measures is determined by the ideal level of lean body mass:” I encourage the authors to delete this.  The abstract is not where the literature is done. 

Line 33 “This manuscript is a continuation of the presentation of the results of the study,…:”  What does this mean and of what value is it to the reader?

Line 90-91 “The results of this study will provide unique information regarding the anthropometric profiles of football players:” I’d delete this.  State it in the Discussion or Conclusion in the present tense, i.e., “…this study provides…”

Lines 97-99: Make these sentences part of the prior paragraph.  Or remove it totally and just go with lines 105-106. Stating it twice is not needed.

Lines 131: Seems strange to have reference numbers at the end of the bullet point.  Better to state this in lines 124-125 as follows, “Measurements were made at the following macrocycle timepoints as defined previously [36,37]”

Line 169 “…parameters were discovered:”  I think the authors mean determined or calculated.

Lines 182-185 “While qualitative data were investigated using percentage notation, quantitative data were represented by mean values and standard deviations (XS). Measurements of the components of total body composition were employed in the study. The comparative value utilized in the statistical tests presented was based on the results.”  What does this mean?  Does it really tell the reader anything more than the Intro and Methods have already informed?  Instead, to the prior paragraph (line 181), it could be simply stated that “The data were summarized as the mean and standard deviation.”

Lines 187: Do these t tests apply to the physical characteristics of the two groups? Or other comparisons?

Lines 195-200: Specifically, what type of ANOVA was used? Presumably 3-way for the number of factors displayed in the graphs.  Please explain as previously requested.

Lines 278-286: The authors start with no known values available for defining where/when REDS could develop.  Later the authors go on to discuss optimal body composition.  What is optimal for football?

Lines 297-303: I suggest making this paragraph the 2nd paragraph.  The first paragraph of the Discussion (which still needs work and has redundancy) lays out the general importance for track body composition.  The next logical communication would be why this study was done.  Then get into the factors that affect and explain the observations in this study, i.e., age, etc.

Lines 304-306 “…younger athletes are characterized by greater fluctuations in the composition of body mass over the macrocycle, it is worth noting that these changes are more intense…:” Are these results or opinions based on the literature?  If the latter, what evidence is there of this in men 18-25 vs 26-31?  The data presented by the authors show differences between age groups but no difference in change/fluctuation.  In other words, fluctuations are not different because the lines in the graphs are parallel indicating no interaction with age over the season.  Further, no data are presented on change in body mass or stature (height), which is what most clinicians would use as the index for growth.

Lines 428-429 “A strong point… and the lack of similar studies in the current scientific literature:” There are quite a few studies on body composition changes over the football season, such as the ones the authors added at the end of the reference list.  Specify how this study is unique.  Because of the focus on segmental or limb comparisons of the composition?

Lines 431-439: Several issues here:

·        No mention is given regarding limb comparison and changes.  What are the implications for these (novel?) findings?

·        What is optimal?  Particularly for lean mass?  Is more mass better? Or is a higher percentage better for a given mass?   

Tables 1 and 2

Both have improved with added detail and consistent decimal places for values.  Check the spelling of median in the footnote for both Tables.

Referencing

Is the journal requirements for referencing being followed?  According to authors instructions “References must be numbered in order of appearance in the text.”  The manuscript follows this through lines 96, with citations in sequence, 1 through 14.  However, in the very next section, they jump to [36,37].  At the end, the authors introduce references 20 and 21. Please thoroughly check the citation order.

Much work needed.  See specific suggestions.

Author Response

Response to Reviewer 1

Thank you so much for your time and insightful assessment. I will respond to your comments below.

Comment 1. The abstract is improved but still needs much work.  For example, nothing is stated about the body composition importance in football.  It isn’t until the 10th line of the abstract that football is mentioned.

The abstract has been revised as suggested, the discipline of football has been clarified at the beginning.

Comment 2. The authors did not address my prior concern re what type of ANOVA was used such as whether a 3-way ANOVA was used since they have three factors, age group by time by limb comparisons.  If not 3-way, why?  The description and/or justification is needed.

Statistical methods were supplemented with the following excerpt:

Three-factor analysis of variance (ANOVA) was used to analyze the composite data in a study of anthropometric profiling and segmental changes in body composition of pro-fessional football players in relation to age during the training macrocycle. This allowed us to understand whether there were statistically significant differences between groups with respect to various factors and interactions between them. In this particular study, a three-factor ANOVA was used due to three significant factors:

Age group: Football players' anthropometric values and body composition can change with age due to the body's growth, maturation, and aging processes. Analysis of age groups identified possible trends or differences in these values across age groups.

Timing (phases of the training macrocycle): There are different phases in the mac-rocycle of training of professional football players, such as the preparatory period, the competitive period, and the transition period. Anthropometric values and body compo-sition may change depending on the type of training and intensity in each phase. Time factor analysis determined whether there are changes in the body composition of football players in different phases of the macrocycle.

Limb comparison: Segmental body composition refers to different parts of the body, such as the upper and lower limbs and the body. Differences in body composition be-tween different body parts can be important in terms of a player's performance, strength, and function. The limb comparison factor allowed us to assess whether there are dif-ferences in body composition between different body parts.

A three-factor ANOVA allowed us to analyze the effect of each factor (age group, time, limb comparison) on variables related to anthropometry and body composition of professional football players.

Comment 3. “Optimal body composition” continues to be expressed but there is no indication of what optimal is for football.  Isn’t the role of this study to help establish what optimal is or the range for optimal? And what should be assessed (limbs)?

There is no clear-cut "optimal" body composition in football, as physical requirements can vary depending on the position on the field and the style of play of a given team. It is worth noting that the final body composition that will be effective in football can vary depending on the team's tactics, style of play, and the coach's preferences. It is crucial that players are in good physical condition, have versatile skills, and are able to adapt to the demands of their role on the field. The purpose of this study is to identify the changes that occur throughout the season and to show whether the age aspect is relevant in this case.

Comment 4. Lines 21-22 “…anthropometric profiling of elite athletes is a useful method for determining their skills, strengths, and weaknesses…:” Does this profiling truly tell us the skills etc. of the athlete? Or does it help us track underlining factors that contribute to performance?

Athlete profiling, also known as "physical profiling" or "athlete profile analysis," is the process of assessing and analyzing an athlete's physical characteristics, technical skills, tactical, psychological, and other aspects related to performance in a given sport. It is a comprehensive approach to understanding an athlete's strengths and areas for improvement in order to reach his or her full potential and to tailor training, tactics, and development plans to meet individual needs. Athlete profiling aims to create personalized training plans and strategies that are tailored to each athlete's individual abilities and potential. This makes it possible to maximize talent and achieve better sports results.

Comment 5. Line 23 “…one of the finest tools available to coaches…:” Remove the superlative “finest.”  Keep your objectivity.  And to be honest, I don’t know that many sport coaches that use body composition instruments.  At least in the US.  Maybe the strength and conditioning coaches.

Corrected as noted. In Poland, body composition analysis is one of the regular elements of evaluating a football player's performance.

Comment 6. Lines 23-24 “…to track body composition for performance and health is the body composition assessment:”  The redundancy of body composition makes this nonsensical.  How about “…to track correlates of performance and health is routine body composition assessment?”

Thank you very much for your comment. Changed as suggested.

Comment 7. Lines 24-26 “According to a study, an excessive amount of body fat significantly impacts performance, power-to-weight ratio, and energy expenditure. However, the importance of individual fitness measures is determined by the ideal level of lean body mass:” I encourage the authors to delete this.  The abstract is not where the literature is done.

Corrected according to the reviewer's comments.

Comment 8. Line 33 “This manuscript is a continuation of the presentation of the results of the study,…:”  What does this mean and of what value is it to the reader?

This notation in this section of the manuscript was introduced as suggested by another reviewer.

Comment 9. Line 90-91 “The results of this study will provide unique information regarding the anthropometric profiles of football players:” I’d delete this.  State it in the Discussion or Conclusion in the present tense, i.e., “…this study provides…”

Corrected according to the reviewer's comments.

Comment 10. Lines 97-99: Make these sentences part of the prior paragraph.  Or remove it totally and just go with lines 105-106. Stating it twice is not needed.

Corrected according to the reviewer's comments.

Comment 11. Lines 131: Seems strange to have reference numbers at the end of the bullet point. Better to state this in lines 124-125 as follows, “Measurements were made at the following macrocycle timepoints as defined previously [36,37]”

Corrected according to the reviewer's comments.

Comment 12. Line 169 “…parameters were discovered:”  I think the authors mean determined or calculated.

I am very sorry for the mistake. Correction.

Comment 13. Lines 182-185 “While qualitative data were investigated using percentage notation, quantitative data were represented by mean values and standard deviations (XS). Measurements of the components of total body composition were employed in the study. The comparative value utilized in the statistical tests presented was based on the results.”  What does this mean?  Does it really tell the reader anything more than the Intro and Methods have already informed?  Instead, to the prior paragraph (line 181), it could be simply stated that “The data were summarized as the mean and standard deviation.”

Corrected according to the reviewer's comments.

Comment 14. Lines 187: Do these t tests apply to the physical characteristics of the two groups? Or other comparisons?

To answer your question: yes, these tests are applicable to comparing the physical characteristics of two groups, provided that the relevant assumptions are met and those in our study were met. As for physical characteristics, you could use the Student's t-test to compare the averages in two different population groups, assuming that the data have a normal distribution in both groups.

Comment 15. Lines 195-200: Specifically, what type of ANOVA was used? Presumably 3-way for the number of factors displayed in the graphs.  Please explain as previously requested.

Described above.

Comment 16. Lines 278-286: The authors start with no known values available for defining where/when REDS could develop.  Later the authors go on to discuss optimal body composition.  What is optimal for football?

Statistical methods were supplemented with the following excerpt:

The optimal body fat content in soccer, which helps prevent RED-S, can vary depending on individual physical characteristics, training intensity, and other factors. Nevertheless, maintaining a healthy and balanced energy balance is key. The key to preventing RED-S in football is a balanced approach to training, nutrition, and recovery that allows athletes to maintain proper energy balance and achieve optimal body composition.

Comment 16. Lines 297-303: I suggest making this paragraph the 2nd paragraph.  The first paragraph of the Discussion (which still needs work and has redundancy) lays out the general importance for track body composition.  The next logical communication would be why this study was done.  Then get into the factors that affect and explain the observations in this study, i.e., age, etc.

Corrected according to the reviewer's comments.

Comment 17. Lines 304-306 “…younger athletes are characterized by greater fluctuations in the composition of body mass over the macrocycle, it is worth noting that these changes are more intense…:” Are these results or opinions based on the literature?  If the latter, what evidence is there of this in men 18-25 vs 26-31?  The data presented by the authors show differences between age groups but no difference in change/fluctuation.  In other words, fluctuations are not different because the lines in the graphs are parallel indicating no interaction with age over the season.  Further, no data are presented on change in body mass or stature (height), which is what most clinicians would use as the index for growth.

Of course, I agree with your comment, corrected in the text. Thank you for your comment.

Comment 18. Lines 428-429 “A strong point… and the lack of similar studies in the current scientific literature:” There are quite a few studies on body composition changes over the football season, such as the ones the authors added at the end of the reference list.  Specify how this study is unique.  Because of the focus on segmental or limb comparisons of the composition?

Clarified as suggested.

Comment 19. Lines 431-439: Several issues here:

  • No mention is given regarding limb comparison and changes. What are the implications for these (novel?) findings?

  • What is optimal? Particularly for lean mass?  Is more mass better? Or is a higher percentage better for a given mass?  

“Changes in the content of lean body mass and fat mass in the age groups were shown, in addition, there was a disproportion in the content of body composition components. Understanding how muscle and fat mass are distributed in the extremities can help coaches adjust training and exercise to improve specific skills. Differences in muscle mass between limbs can lead to muscle imbalances, which in turn can increase the risk of injury. The results of these studies can identify possible imbalances between body sides. This information can be useful for profiling individual players, which can help determine their role on the field and appropriate tactics.”

More muscle mass is not necessarily always better for football players, as it depends on many factors, such as the player's role on the field, the team's style of play, position, individual aptitude, and training goal. There are many aspects to consider:

Different positions in the field require different physical skills. For example, defenders may focus on strength and endurance, while wingers on speed and agility.

Players with too much muscle mass may lose out on speed and agility. With too much mass, they may find it difficult to make quick changes of direction and achieve high speeds.

Greater muscle mass can affect the economy of movement and energy consumption. This does not always automatically mean better performance over longer distances.

Both too much muscle mass and too little muscle mass can increase the risk of injury, especially if muscles and joints are not properly prepared for intense training.

Too much muscle mass can make it difficult to maintain balance and coordinate movements.

A team's style of play can influence which physical attributes are prioritized. Some teams prefer speed and technique, while others may rely on physical strength.

Ultimately, the key is to achieve the right balance between muscle mass and football skills. Therefore, an individualized approach to training that takes into account each player's goals and requirements is necessary.

Comment 20. Tables 1 and 2

Both have improved with added detail and consistent decimal places for values.  Check the spelling of median in the footnote for both Tables.

Corrected according to the reviewer's comments.

Comment 21. Is the journal requirements for referencing being followed?  According to authors instructions “References must be numbered in order of appearance in the text.”  The manuscript follows this through lines 96, with citations in sequence, 1 through 14.  However, in the very next section, they jump to [36,37].  At the end, the authors introduce references 20 and 21. Please thoroughly check the citation order.

The reference has been corrected in accordance with the guidelines.

Thank you for your help. Your guidance is invaluable.

Kind regards,

Authors.

Reviewer 2 Report

The authors did address all the major concerns that were previously raised, and with this the manuscript improved. The re-submmitted version of this paper is acceptable for publication. 

However, one point (multiple repeats) still needs to be taken care of:

Line 401 I still notice the use of 'author's paper' which is a very unspecific and unclear phrasing. Perhaps it should read like: 'Decreased muscle mass caused by a lack of training stimulus can result in decreased strength and endurance, and thus an  increased risk of injury when numerous and intense training units are reintroduced during the preparation period [35], which also showed an increased in lean body mass and fat mass in all body segments analyzed.'

Please also correct the use of 'author's study' in Lines 353 and 376.

Author Response

Response to Reviewer 2

Thank you very much for your time and positive evaluation of the survey.

Comment 1. Line 401 I still notice the use of 'author's paper' which is a very unspecific and unclear phrasing. Perhaps it should read like: 'Decreased muscle mass caused by a lack of training stimulus can result in decreased strength and endurance, and thus an  increased risk of injury when numerous and intense training units are reintroduced during the preparation period [35], which also showed an increased in lean body mass and fat mass in all body segments analyzed.'

Please also correct the use of 'author's study' in Lines 353 and 376.

Changed in accordance with the reviewer's comment. In addition, the entire study was revised and the wording "author's study" was changed.

Thank you for your help. Your guidance is invaluable.

Kind regards,

Authors.

Round 3

Reviewer 1 Report

Lines 21-22 "...therefore anthropometric profiling of elite football players is a useful method of determining their skills, strengths, and weaknesses to develop..."  Based on the authors' response, anthropometric profiling seems to be a component of the comprehensive profiling.  I suggest restating it like this (addition in caps): "...therefore anthropometric profiling of elite football players is [DELETE a useful method] USEFUL AS A PART of determining their skills, strengths, and weaknesses to develop..."  Anthropometry in and of itself doesn't assess skills, strength, or weakness but it (body composition) is an underlying factor for skills, etc.

None.

Author Response

Thank you very much for your time and positive evaluation of the survey.

Comment 1. Lines 21-22 "...therefore anthropometric profiling of elite football players is a useful method of determining their skills, strengths, and weaknesses to develop..."  Based on the authors' response, anthropometric profiling seems to be a component of the comprehensive profiling.  I suggest restating it like this (addition in caps): "...therefore anthropometric profiling of elite football players is [DELETE a useful method] USEFUL AS A PART of determining their skills, strengths, and weaknesses to develop..."  Anthropometry in and of itself doesn't assess skills, strength, or weakness but it (body composition) is an underlying factor for skills, etc.

Changed according to reviewer's comment. Thank you very much for your valuable suggestions.

Thank you for your help. Your guidance is invaluable.

Kind regards,

Authors.
